# Mapping Tumor Spheroid Mechanics in Dependence of 3D Microenvironment Stiffness and Degradability by Brillouin Microscopy

**DOI:** 10.3390/cancers13215549

**Published:** 2021-11-05

**Authors:** Vaibhav Mahajan, Timon Beck, Paulina Gregorczyk, André Ruland, Simon Alberti, Jochen Guck, Carsten Werner, Raimund Schlüßler, Anna Verena Taubenberger

**Affiliations:** 1Center for Molecular and Cellular Bioengineering (CMCB), BIOTEC, Technische Universitaet Dresden, 01307 Dresden, Germany; vaibhav.mahajan@tu-dresden.de (V.M.); timon.beck@mpl.mpg.de (T.B.); 299749@uwr.edu.pl (P.G.); simon.alberti@tu-dresden.de (S.A.); raimund.schluessler@tu-dresden.de (R.S.); 2Max Planck Institute for the Science of Light & Max-Planck-Zentrum für Physik und Medizin, Staudtstr. 2, 91058 Erlangen, Germany; jochen.guck@mpl.mpg.de; 3Max Bergmann Center, Leibniz Institute of Polymer Research Dresden, 01069 Dresden, Germany; ruland@ipfdd.de (A.R.); carsten.werner@tu-dresden.de (C.W.)

**Keywords:** cell mechanics, tumor microenvironment, tumor spheroid, 3D culture, compression, atomic force microscopy, Brillouin microscopy, confinement

## Abstract

**Simple Summary:**

Little is known about how cancer cells adapt their mechanical properties in complex 3D microenvironments. Here we generated different types of tumor spheroids within compliant or stiff hydrogels. We then quantitatively mapped the mechanical properties of these spheroids in situ using Brillouin microscopy. Maps acquired for tumor spheroids grown within stiffer hydrogels showed elevated Brillouin shifts, hence spheroids became “stiffer” compared to the ones cultured within compliant gels. The spheroid’s mechanical properties were modulated by various microenvironment properties including matrix stiffness and degradability and the resultant compressive stress but also depending on whether single cells or cell aggregates were analyzed. Moreover, spheroids generated from a panel of invasive breast, prostate and pancreatic cancer cell lines within degradable stiff hydrogels became stiffer and at the same time, less invasive compared to those in compliant hydrogels. Taken together, our findings contribute to a better understanding of the interplay between cancer cells and their microenvironment, which is relevant to better understand cancer progression.

**Abstract:**

Altered biophysical properties of cancer cells and of their microenvironment contribute to cancer progression. While the relationship between microenvironmental stiffness and cancer cell mechanical properties and responses has been previously studied using two-dimensional (2D) systems, much less is known about it in a physiologically more relevant 3D context and in particular for multicellular systems. To investigate the influence of microenvironment stiffness on tumor spheroid mechanics, we first generated MCF-7 tumor spheroids within matrix metalloproteinase (MMP)-degradable 3D polyethylene glycol (PEG)-heparin hydrogels, where spheroids showed reduced growth in stiffer hydrogels. We then quantitatively mapped the mechanical properties of tumor spheroids in situ using Brillouin microscopy. Maps acquired for tumor spheroids grown within stiff hydrogels showed elevated Brillouin frequency shifts (hence increased longitudinal elastic moduli) with increasing hydrogel stiffness. Maps furthermore revealed spatial variations of the mechanical properties across the spheroids’ cross-sections. When hydrogel degradability was blocked, comparable Brillouin frequency shifts of the MCF-7 spheroids were found in both compliant and stiff hydrogels, along with similar levels of growth-induced compressive stress. Under low compressive stress, single cells or free multicellular aggregates showed consistently lower Brillouin frequency shifts compared to spheroids growing within hydrogels. Thus, the spheroids’ mechanical properties were modulated by matrix stiffness and degradability as well as multicellularity, and also to the associated level of compressive stress felt by tumor spheroids. Spheroids generated from a panel of invasive breast, prostate and pancreatic cancer cell lines within degradable stiff hydrogels, showed higher Brillouin frequency shifts and less cell invasion compared to those in compliant hydrogels. Taken together, our findings contribute to a better understanding of the interplay between cancer cells and microenvironment mechanics and degradability, which is relevant to better understand cancer progression.

## 1. Introduction

Tumor cells reside within a complex microenvironment that regulates essential cellular functions and also plays an important role in tumor progression and metastasis [1,2]. The underlying tumor-microenvironment interactions include biochemical but also mechanical cues that are sensed by tumor cells. For several types of cancer, including breast [3,4], colon [5], prostate [6] and pancreatic cancer [7], changes in the mechanical properties of tumor tissue have been reported when compared to normal tissue. Tissue stiffening is typically associated with stiffer and denser extracellular matrix (ECM), which is thought to be primarily caused by enhanced matrix deposition and remodeling by cancer-associated fibroblasts [8]. The mechanical abnormalities do not only include tissue stiffening but also a build-up of compressive stress that expanding tumors are exposed to in particular when growing in a less pliable environment [9]. In addition, interstitial fluid pressure increases due to leaky blood vessels [10]. Thus, cancer cells have to survive and proliferate in a mechanically confining environment and—at the onset of metastasis—also invade through a stiffer and denser matrix.

Since increased ECM stiffness has been correlated with tumor aggressiveness [11,12,13], it is therefore important to better understand the response of the tumor cells to a stiffened microenvironment. Over the past years, several in vitro studies have shown that tumor cell growth and invasion are regulated by microenvironment stiffness [12]. Using a 2D gel, it has previously been demonstrated that the mechanical properties of the matrix on which cells are cultured, affects the morphology and proliferation of cells [14]. Mechanical properties of the environment have also been recognized to affect cell migration, differentiation, and cell volume [15,16,17,18]. Moreover, cell stiffness was shown to be modulated by matrix stiffness [19,20]. However, most studies on the mechanical interactions between cells and their microenvironment have been performed in 2D [21] or using single cells in a 3D environment [22]. In the context of multicellular systems, the relationship between microenvironmental stiffness and tumor cell mechanics and invasion has been much less explored despite its importance to cancer progression [23,24].

Multicellular 3D cell culture models such as tumor spheroids recapitulate relevant features of epithelial tumors, including more realistic cell-cell contacts, proliferation rates, diffusion gradients and drug responses [25,26,27]. We have recently established a 3D tumor spheroids culture system based on PEG (polyethylene glycol)-heparin, which allows studying tumor spheroid growth within mechanically defined microenvironments. Thereby spheroids were generated from single cells proliferating within biohybrid PEG heparin gels. Advantageously, these gels allow for independent tuning of biochemical factors/ligand density and mechanical properties. In addition, synthetic hydrogels permit tailored modification of matrix parameters such as adhesion sites (e.g., RGD peptides) and degradability through the incorporation of metalloproteinase (MMP) cleavage sites [28,29]. Using this experimental model, we have previously shown that spheroids from luminal epithelial breast cancer cell lines are exposed to compressive stress and stiffen when grown in stiffer hydrogels [30]. Taking the spheroids out of the strained hydrogel for studying their mechanics by AFM, however, leads to a loss of the compressive stress [10], and thereby might affect the spheroids’ mechanical phenotype. Thus, we wished to employ a non-invasive technique which would allow us to analyze the mechanical phenotype of the spheroids within their hydrogel environment in situ.

For this reason, we set out here to investigate the mechanical properties of tumor spheroids in situ using Brillouin microscopy (BM) [31]. It is a label-free, noncontact method with subcellular optical resolution [32,33]. BM is based on Brillouin scattering, the inelastic scattering of light by propagating pressure waves in a material. Such collective molecular movements give rise to a frequency shift of the scattered light of a few GHz in biological samples. The resultant Brillouin frequency shift can be used to calculate the elastic (longitudinal) modulus of the sample if its density and refractive index are known. BM is currently being used and developed for diverse applications in biomedical research such as ophthalmological pathologies and basic research such as mouse embryogenesis, zebrafish development and regeneration, and cell biology [34,35,36]. Recent work has indicated that for most common biological samples the Brillouin shift can be used as a proxy for the sample’s mechanical properties [31,37]. Thus, BM presents an ideal tool to study the mechanical phenotype of spheroids within hydrogels. So far, only a handful of studies have studied tumor tissue and spheroids with BM. For melanoma tissues, a distinct change in Brillouin frequency shift between healthy and malignant tissue was shown in the porcine model [38]. Two recent studies have analyzed the mechanical properties of colorectal tumor spheroids [39] and ovarian cancer cells by BM [40] and the effect of chemotherapeutic drug treatments. The effect of a stiffened microenvironment on spheroid mechanics has not been investigated by BM yet.

We report here that different types of tumor spheroids growing in stiff degradable hydrogels are characterized by a larger Brillouin frequency shift, hence increased elastic moduli, as compared to spheroids in compliant degradable hydrogels. This indicates that the spheroids adapted their mechanical properties to their microenvironment. Systematically comparing different culture systems where differences in compressive stress occur, e.g., after blocking hydrogel degradation, probing single cells or free spheroids, revealed a positive relationship of increasing Brillouin frequency shifts and compressive stress. Spheroids formed from invasive breast, prostate and pancreatic cancer cells formed in stiffer degradable hydrogels, showed higher Brillouin shifts and were less invasive compared to those in compliant degradable hydrogels. Thus, we find that the spheroids’ mechanical phenotype and cell invasiveness are modulated by matrix stiffness, matrix degradability, multicellularity and concomitant changes in compressive stress levels. Together, our results bring new insights into the mechanical interactions between tumor cells and their microenvironment, which is relevant to better understand cancer progression.

## 2. Materials and Methods

### 2.1. Cell Culture

MCF-7, MDA-MB-231 and PC3 cells were obtained from the “Deutsche Stammsammlung für Mikroorganismen und Zellkulturen” (DSMZ). PANC1 cells were kindly provided. MCF-7 cells were maintained in RPMI 1640 (Thermofisher, Waltham, MA, USA) supplemented with 10% heat-inactivated fetal bovine serum (FBS), 1 mM sodium pyruvate, MEM non-essential amino acids, 10 µg/mL human insulin (Sigma Aldrich, St. Louis, MO, USA) and penicillin (100 U/mL) -streptomycin (100 µg/mL). MDA-MB-231, PANC1 and PC-3 cells were cultured in DMEM/F-12, DMEM-GlutaMAX and RPMI 1640 respectively, supplemented with 10% fetal bovine serum (FBS) and penicillin (100 U/mL) -streptomycin (100 µg/mL). Before starting 3D cultures, cells were grown in standard T25 culture flasks and sub-passaged 2–3 per week. For cell detachment, TrypLE^®^ was used. All cell culture reagents were from Thermofisher unless otherwise stated.

### 2.2. Preparation of PEG-Heparin 3D Hydrogel Cultures

PEG and heparin-maleimide precursors were synthesized as previously described [41]. Cells were detached from tissue culture flasks with TrypLE^®^, resuspended in cell culture medium, centrifuged down, and resuspended in PBS. Then they were mixed with the freshly prepared heparin-maleimide (MW~15000) solution and RGD peptide (final 3 µM), at a final density of 5.6 × 10^5^ cells/mL and a final heparin-maleimide concentration of 1.5 mM. For a molar ratio, PEG/heparin-maleimide of gamma = 1.5, PEG precursors (MW~16000) were reconstituted with PBS at a concentration of 2.25 mM and placed into an ultrasonic bath (Merck) for 20 s (medium intensity). Lower stiffness gels were prepared by diluting the stock solution of PEG precursors accordingly (1:2 for gamma = 0.75). To prepare PEG-heparin hydrogel droplets, heparin-cell suspension was mixed in a chilled microtube with the same amount of PEG solution using a low binding pipette tip. Then, a 25 µL drop of the PEG-heparin-cell mix was pipetted onto hydrophobic, Sigmacote^®^ (Sigma Aldrich)-treated glass slides (VWR). Gel polymerization started immediately, and stable hydrogels were obtained within 1–2 min. Hydrogels were gently detached from the glass surface after 4 min using a razor blade and transferred into a 24 multi-well plate supplemented with 1 mL cell culture medium. Cell culture medium was exchanged every other day.

### 2.3. Quantification of Hydrogel Mechanical Properties by AFM

After their preparation, hydrogels were immersed in PBS and stored at 4 °C until mechanical characterization on the following day. After equilibrating them at RT for 1 h, gels were mounted using CellTak (Thermofisher, Waltham, MA, USA) onto glass object slides (VWR, Radnor, PA, USA). A Nanowizard I or IV (JPK Instruments, Berlin, Germany) was used to probe the gel stiffness. Cantilevers (arrow T1, Nanoworld, Neuchâtel, Switzerland) that had been modified with polystyrene beads of 10 μm diameter (Microparticles GmbH, Berlin, Germany) using epoxy glue (Araldite, Huntsman Corporation, The Woodlands, TX, USA), were calibrated using the thermal noise method implemented in the AFM software (Version 6.1.159, NanoWizard Control Software, JPK Instruments/Bruker, Berlin, Germany). Hydrogels were probed at RT in PBS using a speed of 5 μm/sec and a relative force setpoint ranging from 2.5 to 4.0 nN in order to obtain comparable indentation depths (0.5–1 µm). Force distance curves were processed using the JPK data processing software (Version 6.1.159, NanoWizard Control Software, JPK Instruments/Bruker, Berlin, Germany). Indentation parts of the force curves (approximately 1.5 μm indentation depth) were fitted using the Hertz/Sneddon model for a spherical indenter, assuming a Poisson ratio of 0.5 [42,43].

### 2.4. Morphometric Analysis of Tumor Spheroids In Situ and Ex Situ

On day 14, cultures were fixed with 4% *formaldehyde/PBS* for 45 min, followed by a 30 min permeabilization step in 0.2% Triton-X100. Spheroids were stained for 2–4 h with 5 µg/mL DAPI and 0.2 µg/mL Phalloidin-TRITC in 2% BSA/PBS. Then hydrogels were immersed at least for 1 h in PBS. Spheroids were imaged with a LSM700 confocal microscope using a 40× objective (Zeiss C-Apochromat, White Plains, NY, USA). Quantitative analysis of cell morphology was done using FIJI. Briefly, a threshold was applied to the fluorescent images (red channel, F-actin), which were then transformed into binary pictures, and shape factor (defined as 4 × π × area/(perimeter)^2^) and area were calculated.

### 2.5. Brillouin Microscopy

Brillouin maps were acquired using a custom-built confocal Brillouin microscope employing a two-stage VIPA spectrometer as previously described in [44,45]. The setup features a frequency-modulated diode laser with a wavelength of 780 nm whose frequency is stabilized to the D2 transition of Rubidium 85. To suppress background light created by amplified spontaneous emission, a Bragg grating and a Fabry–Pérot interferometer in a two-pass configuration are used. All images were obtained with a 20×/0.5 air objective and the sample temperature was controlled to 37 °C by a petri dish heater (JPK BioAFM, Berlin, Germany).

### 2.6. Statistical Analysis

GraphPad Prism was used to plot data and to perform statistical tests. A two-tailed significance level of 5% was considered statistically significant (*p* < 0.05). As indicated in the figure legends, for pairwise comparisons a Mann–Whitney (non-parametric) test was used since most datasets were not normally distributed. In the case of more than two groups, a Kruskal–Wallis (non-parametric) with multiple comparisons (Tukey) was chosen. To test for normality, a D’Agostino–Pearson omnibus normality test was performed.

## 3. Results

### 3.1. Stiff Degradable Hydrogels Reduce the Growth of MCF-7 Spheroids

To study how microenvironment stiffness affects the growth and mechanics of tumor spheroids, we generated MCF-7 (luminal breast cancer cell line) spheroids by embedding single cells into PEG-heparin hydrogels and letting them grow to multicellular aggregates (Figure 1A). Hydrogel degradability could be systematically modified by using PEG precursors with incorporated MMP cleavage sites. The elastic modulus of degradable or non-degradable hydrogels could be easily adjusted to a range from 1–2 kPa (for compliant gels) to about 15–20 kPa (for stiff gels), encompassing stiffness values previously reported for human breast tumor specimens [46] (Figure 1B). During the culture period of 14 days, tumor spheroids of up to 160 µm in diameter formed (Figure 1C). We observed that spheroids showed significantly reduced cross-sectional areas in the stiff hydrogel as compared to compliant hydrogels (Figure 1D). Further analyzing the morphology of the tumor spheroids, we found that in the stiff hydrogels, spheroids displayed a greater cell density as quantified by the number of cells per unit of cross-sectional area (Figure 1E). Overly similar shapes of the spheroids were seen (Figure 1F). Thus, stiff hydrogels reduced MCF-7 spheroids’ growth and caused a higher degree of spheroid compaction. These results are in line with our previous work, where spheroid growth in stiff versus compliant degradable hydrogels was associated with increased compaction along with increased compressive stress as quantitated by elastic stress sensors [30]. In this way, radial stresses could be determined by the deformation of fluorescent elastic polyacrylamide beads (Figure 1G). Using the same elastic beads as pressure sensors, we estimated here the compressive stress acting on grown tumor spheroids (Appendix A). Plotting radial stresses over the respective spheroid radii reached after a culture period of 14 days, revealed an inverse relationship between both (Figure 1H). This suggests that spheroids adapted their growth to the level of compressive stress varying with matrix stiffness and degradability (Figure 1H).

### 3.2. MCF-7 Spheroids in a Stiff Degradable Gel Show an Increased Brillouin Frequency Shift

After seeing a difference in spheroid growth and compaction in response to hydrogel stiffness, we set out to investigate how matrix stiffness would affect the mechanical phenotype of the spheroids in situ. We employed BM (Figure 2A) to quantitatively assess the mechanical properties of spheroids grown in hydrogels of different stiffness. From spectra acquired with our Brillouin microscopy setup (Figure 2B) we derived the Brillouin frequency shift for each x-y position of the map, which is related to the longitudinal modulus and therefore is used here as a proxy for the sample’s elastic properties. As seen in the representative Brillouin maps (Figure 2C), clearly distinct Brillouin frequency shifts of hydrogel and spheroid were obtained. Averaged line profiles over the spheroid width showed increased Brillouin frequency shifts towards the spheroid center (Figure 2D). Segmenting for the spheroid or hydrogel region (Figure 2E) allowed us then to separately analyze the mechanical properties of hydrogel and spheroid. In line with the AFM analysis (Figure 1B), the Brillouin frequency shifts measured for stiff hydrogels were significantly increased compared to compliant hydrogels, hence indicating a higher longitudinal modulus (Figure 2F). Regarding the spheroids’ mechanical properties, we detected significantly increased median Brillouin frequency shifts and therefore higher longitudinal moduli for spheroids grown in stiff compared to compliant degradable hydrogels (Figure 2G and Appendix A). For the Brillouin line width (relating to the material’s viscosity [32]) comparable values between compliant and stiff hydrogel cultures were obtained (Appendix A). In sum, these results show that hydrogel stiffness affected not only spheroid growth, but also the mechanical properties of MCF-7 spheroids, which apparently became stiffer in the stiffer hydrogels.

### 3.3. Interfering with Hydrogel Degradation Affects the Brillouin Frequency Shifts of MCF-7 Spheroids

In order to investigate whether the seen changes in Brillouin frequency shift were a response of compressive stress acting on the tumor spheroids during growth, we next set out to manipulate the levels of compressive stress on growing tumor spheroids. Firstly, we tested how microenvironment degradability would affect the spheroid’s mechanical properties since our bead pressure sensors had indicated that matrix degradability had a large effect on the growth-induced compressive stress (Figure 1H). Therefore, we probed spheroid mechanics by BM while blocking hydrogel degradability using two different ways: 1. By forming MCF-7 spheroids in non-degradable gels (Figure 3A) and by 2. growing spheroids within degradable hydrogels in presence of an MMP inhibitor (GM6001) (Figure 3C). Analogously to degradable hydrogels (Figure 1D), spheroid sizes were significantly reduced in stiff versus compliant hydrogels under both degradation-blocking conditions (Figure 3B,D), although for non-degradable hydrogels generally smaller spheroids were observed compared to degradable hydrogels (see green line Figure 3B,D). For spheroids growing in non-degradable compliant and stiff hydrogels, overly round shapes and similar cell densities were determined (Figure 3B and Appendix A). In contrast, MMPi treated spheroids grown in stiff degradable hydrogels showed a greater cell density compared to compliant gels (Figure 3D). With regards to their mechanical properties, for spheroids in non-degradable hydrogels, mean Brillouin frequency shifts gradually increased towards the spheroid cores (Appendix A). For both conditions where matrix degradation was blocked, similar Brillouin frequency shifts were obtained for spheroids grown in compliant and stiff hydrogels (Figure 3E,F and Appendix A). This was different from the spheroids grown in degradable hydrogels (Figure 2G) and is in line with the hypothesis that the adaption of the spheroid’s mechanical properties is modulated by compressive stress.

### 3.4. Single Cells within Hydrogels Display Low Brillouin Frequency Shifts

To test this idea further, we next investigated the mechanical properties of single MCF-7 cells in hydrogels, since these should be initially not exposed to significant levels of compressive stress (Figure 4A). Indeed, significantly decreased Brillouin frequency shifts were found for single MCF-7 cells compared to matrix-embedded spheroids (Figure 4B, comparison to non-degradable: *p* = 0.001), with a trend of increasing Brillouin frequency shifts with hydrogel stiffness. Moreover, unconfined spheroids were generated by MCF-7 cell aggregation in low-adhesive U-well plates (Figure 4C,D). Spheroids displayed comparable densities compared to matrix embedded spheroids (*p* = 0.73 compared to non-degradable compliant). BM on unconfined spheroids (Figure 4E), however, revealed higher Brillouin frequency shifts compared to single cells but lower shifts compared to matrix-embedded spheroids (Figure 4F). We next wished to compare Brillouin frequency shifts of the different culture conditions ranking by the level of compressive stress. For matrix embedded spheroids we had already estimated the level of compressive stress actually acting on the spheroids in degradable and non-degradable compliant and stiff gels (Figure 1G,H). For single cells and unconfined spheroids, we could not determine the actual stress but assumed a generally low level of compressive stress. Interestingly, Brillouin frequency shifts increased with the estimated compressive stress level (Figure 4G). Summing up, these observations suggest that the compressive stress felt by the spheroids, which depends not only on matrix stiffness but also its degradability, modulated the spheroid’s mechanical phenotype seen by BM. Given the differences in Brillouin shift between single cells and unconfined spheroids, however, also other factors might affect the mechanical phenotype e.g., multicellularity and the associated cell-cell interactions, as well as cellular interactions with the compliant or stiff matrices.

### 3.5. In Stiff Non-Degradable Hydrogels, Invasive Cancer Cells Show Reduced Spheroid Formation and Brillouin Frequency Shifts

After exploring the effect of microenvironment stiffness and degradability on spheroids from non-invasive MCF-7 cells, we wondered how invasive cell behavior would be coupled to microenvironment and spheroid mechanics. Thus, we embedded MDA-MB-231 (basal breast cancer cell line), PANC1 (pancreatic cancer cell line) and PC3 (prostate cancer cell line) in compliant and stiff non-degradable gels (Figure 5A–C), all of which are invasive cancer cell lines. In contrast to MCF-7 cells, the panel of invasive cell lines failed to form proper multicellular spheroids within stiff non-degradable hydrogels. Only small aggregates consisting of about 2–3 cells per cross-section were observed. In compliant hydrogels, multicellular spheroids formed for all cell lines, although considerable size variations were seen, especially for PANC1 and PC3 cells (Figure 5D–F). While cell density increased for MDA-MB-231 cells in stiff hydrogels, for PC-3 and PANC1 decreased values were seen (Appendix A). Overly, shapes were—as expected due to the lack of hydrogel degradability—rather round as indicated by the shape factors (>0.8) and no invasive events were seen (Figure 5D–F). In contrast to MCF-7 spheroids in non-degradable hydrogels, we observed that the Brillouin frequency shifts for the MDA-MB-231, PANC1 and PC3 cells growing in a stiff gel were lower compared to spheroids in compliant hydrogels (Figure 5G–I). Since we had quantified that all three cell lines only formed very small clusters or even remained as single cells in stiff hydrogels, the relatively low Brillouin frequency shifts in stiff versus compliant hydrogels may be attributed to the lower levels of compressive stress, analogously to what we had observed for single MCF-7 cells (Figure 4).

### 3.6. In Degradable Hydrogels, Invasive Cancer Cells Invade and Have Decreased Brillouin Frequency Shifts

We then set out to explore how these invasive cancer cell lines behaved in a microenvironment that would permit cell invasion. Therefore, we seeded MDA-MB-231, PANC1 and PC3 cells in compliant and stiff degradable hydrogels to generate spheroids. In the compliant degradable hydrogels, MDA-MB-231 cells formed no spheroids as they dispersed through the gel (Figure 6A), which is in line with [47]. Similar to MCF-7 spheroids, PANC1 and PC3 spheroids were larger in compliant hydrogels. At the edge of PANC1 and PC3 spheroids invading cells were seen in compliant hydrogels (Figure 6B,C, white arrows). In contrast, in stiff degradable hydrogels, all three invasive cell lines formed spheroids but showed fewer signs of invasion compared to the compliant degradable hydrogels. The more irregular spheroid edge in compliant hydrogels was in accordance with the decreased shape factors of PANC1 spheroids (Figure 6D–F). A higher cell density within PANC1 but not PC3 spheroids growing in stiff hydrogels was seen when compared to the compliant gels (Appendix A). Interestingly, the Brillouin frequency shifts for spheroids formed from invasive cell lines (Figure 6G) followed a similar tendency as MCF-7 spheroids in degradable hydrogels, since spheroids in stiff hydrogels consistently had a higher frequency shift compared to compliant hydrogels (Figure 6H). Therefore, lower spheroid Brillouin frequency shifts of spheroids in compliant degradable hydrogels were associated with increased invasiveness. Since we suspected that (fluid- or hydrogel-filled) spaces between invading cells might have contributed to lower overall Brillouin frequency shifts, we also compared the top 5% of Brillouin frequency shift values over the spheroids, supposedly corresponding to areas with cellular structures (Appendix A). Additionally, when taking this top 5% into account, we found significantly higher Brillouin frequency shifts for MDA-MB-231 and PC3 cells/spheroids when grown in stiffer hydrogels, while the above-seen changes in PANC1 spheroids were almost leveled out. The higher Brillouin frequency shifts in stiff degradable hydrogels therefore indicate that spheroids adapted their mechanical properties to the stiffened microenvironment, which was associated with decreased invasion and -at least for MDA-MB-231 and PC3 cells- with changes in the mechanical properties of constituent cells.

## 4. Discussion

In the presented study, we have investigated the influence of microenvironment stiffness on tumor spheroid growth, morphology, invasion and mechanical properties using a platform combining engineered tumor microenvironments and BM, an emerging tool in the biomechanics field. To investigate spheroid mechanics in 3D microenvironments of defined stiffness, we have chosen in our study a biohybrid PEG-heparin hydrogel system. Our 3D engineered model incorporates different aspects of mechanical cell interactions, such as growth-induced compressive stress, cell-cell, as well as cell-matrix interactions (by including RGD peptides). Importantly the hydrogel’s mechanical properties can be tuned independently of ligand density. Young’s moduli of the predominantly elastic hydrogels, as characterized by AFM, ranged between 1 and 20 kPa and were therefore within the range of elastic moduli reported for breast tumor specimens [46]. When we segmented the obtained BM maps for the hydrogel area, we found higher Brillouin frequency shifts for stiff compared to compliant hydrogels. Brillouin frequency shift and longitudinal modulus are directly related via parameters including refractive index and density [32]. Recent studies on different biological systems using a combined BM and optical diffraction tomography (ODT) setup have shown that changes in the Brillouin frequency shift are predominantly due to changes in the longitudinal modulus [37]. The more commonly used elastic modulus to describe the mechanical properties of cells is Young’s modulus, which can be directly obtained by AFM indentation tests. Longitudinal modulus and Young’s modulus are related via the Poisson ratio, although it is not straightforward to directly calculate one from the other, since the Poisson ratio is frequency-dependent. Nevertheless, our results showing higher Brillouin frequency shifts for gels of higher Young’s modulus are in line with other studies showing a direct relationship between Brillouin frequency shift and Young’s modulus for different hydrogels, e.g., polyacrylamide [31,47], photo-crosslinked PEG [31] or GelMA [48]. These results further validate the Brillouin frequency shift as a proxy for the elastic (longitudinal) modulus of the sample. One should note here, however, that the Brillouin shift is also impacted by the polymer content and that the correlation with the Youngs modulus might not be observable if the polymer fraction of the hydrogel is too small [49,50] as previously suggested.

While these results were obtained on hydrogels, also various experiments have been conducted in the past to reveal biological properties that influence the Brillouin frequency shift measured for cells. For instance, drugs that interfere with F-actin polymerization were combined with BM, which resulted in decreased Brillouin frequency shifts in different cell types [31,51]. Additionally, higher Brillouin frequency shifts of cells were previously measured with increasing osmolarity of the medium surrounding the cells, a treatment that typically increases Young’s modulus measured by indentation tests [39]. Another positive correlation between AFM indentation tests and Brillouin frequency shifts were recently reported, when fibroblasts growing on polyacrylamide gels were investigated by BM [31]. Similar to previous reports by AFM [52], where Young’s moduli increased with gel stiffness, increased Brillouin frequency shift were found for stiffer hydrogels. Together, these experiments underline the relevance of BM for studies of cell and tissue mechanical properties.

When we compared the spheroids’ mechanical properties within the BM maps, we found increased Brillouin frequency shifts for spheroids in stiff degradable gels. Plotting Brillouin frequency shift of spheroids versus Brillouin frequency shift of hydrogels, a direct relation was seen (Appendix A), which suggests that spheroids adapted their mechanical properties to increasing hydrogel stiffness. To our knowledge, this is the first report showing changes in the mechanical properties of tumor spheroids in response to microenvironment stiffness by BM. The mechanical changes at spheroid level might be explained by different factors. Firstly, this stiffening could be at least partly due to an increased compaction of the spheroids as mirrored by the increased number density of cells in spheroids in stiff hydrogels. Indeed, with the exception of MMPi treated spheroids, higher Brillouin frequency shifts were in most cases associated with increased cellular density of spheroids. Compaction could be due to fewer intercellular spaces and/or reduced individual cell volumes. Considering the seen numbered density increase of approximately 25–30%, it is unlikely to be explained solely by cell volume reduction in stiff hydrogels, since this would require larger hydrostatic pressures in the hundreds of kPa range as previously shown [18]. Thus, it appears more likely that the seen compaction is due to similar-sized cells coming closer to each other, which could also cause an increase in Brillouin frequency shift, since intercellular spaces should result in lower Brillouin frequency shifts. Second, the increased Brillouin frequency shift at spheroid level in stiff hydrogels could be associated with stiffness increase of individual cells or a combination of both, compaction and cell stiffening. Given our map resolution of 2 µm, this allowed us to further dissect intracellular contributions. When we compared the top 5, 10, and 15% of Brillouin frequency shift values measured within the MCF-7 spheroid area (Appendix A), we consistently obtained increased Brillouin frequency shifts for spheroids within stiff hydrogels, which suggests that cells in stiff degradable gels became stiffer. These results are also in line with two previous studies using optical tweezers showing a stiffening response of breast and pancreatic cancer cells to collagen hydrogels of increasing concentrations [23,24]. The adaption to microenvironment stiffness would also be in line with our previous report using AFM [30], showing higher apparent Young’s moduli for spheroids and constituent cells from stiff hydrogel cultures compared to compliant cultures. However, to get access to the spheroid surface or constituent cells in the previous study, spheroids had to be released from the gels. It cannot be excluded though that spheroid isolation might affect the mechanical properties of the spheroids adapting to their new situation. Using BM instead of AFM has the advantage that spheroids could be probed in a non-contact manner and within the context of an intact hydrogel, where elastic stresses exerted by a strained hydrogel matrix remain. Moreover, the measurements are not limited to the spheroid surface. This spatial information is valuable since it appears that the mechanical properties of the spheroids are not homogeneously distributed. Firstly, we find in some spheroids’ areas of lower Brillouin frequency shift, which might be related to small cavities or small lumens within the spheroid. Additionally, we find an increase of the Brillouin frequency shift towards the spheroids’ center, similar to a recent BM study on colorectal tumor spheroids [38]. While increased Brillouin frequency shifts might be explained by an increasing crowdedness within the center, it may also hint at stiffer cells within the inner part of the spheroids. A similar phenomenon was reported for breast cancer spheroids in a recent study, where the mechanical properties of individual MCF10A cells were probed in dependence of their localization within a spheroid using optical tweezers [53]. The study showed that cell stiffness was lowest at the rim and highest within the core of the spheroid [53]. Recently changes in mechanical properties were also seen for single cells residing within hydrogels of different elastic moduli, e.g., for breast cancer cells in collagen gels [22] or alginate gels [45]. Altogether, our results suggest that the observed changes in spheroid mechanical properties occur at different scales, at the single-cell level as well as the level of spheroid.

Under invasive conditions, i.e., invasive cell lines in compliant degradable gels, we have found lower overall Brillouin frequency shifts, which might also indicate decreased cell stiffness of invasive cells or result from a looser structure. However, also for PC-3 cells, where a similar cell density was seen in compliant and stiff degradable hydrogels, a lower Brillouin frequency shift was detected for invasive spheroids in compliant hydrogels. Since we suspected that the Brillouin frequency shift for the invading single cells might have been biased by averaging with the surrounding hydrogel within the scattering volume, we decided to look at the maximum values within each map. In compliant degradable gels, we observed lower Brillouin frequency shifts for invading cells compared to no non-invading cells in stiff hydrogels, which fits the picture of invasive cells having decreased stiffness values [53]. In contrast, a recent study—comparing two cell lines of different metastatic potential—has detected no differences in the Brillouin frequency shift, but instead in the line width (longitudinal loss modulus) [38]. This indicates that BM can potentially detect relevant cell mechanical changes of cancer cells although more work is needed to better understand the structural basis of these changes. The finding of decreased Brillouin frequency shifts correlating with increased invasiveness is also in agreement with multiple studies comparing cancer cells of different invasiveness in 2D cultures or in suspension [22,54,55,56,57]. In our and the above-mentioned optical tweezer study [53], however, we could find changes in cell mechanical properties in response to mechanically altered complex 3D microenvironments.

The finding that so little invasion occurs in stiff degradable hydrogels appears at first sight contradictory to previous work showing greater invasion in stiffer microenvironments [12]. However, the stiffness range and biomaterial used were quite different. Our hydrogel system is characterized by subcellular-sized pores (10–20 nm) and elastic moduli of about 20 kPa for the stiffer hydrogels. Under these conditions, the stiff hydrogels might rather form a barrier to invading cells as also shown by the other group’s work [58]. In addition, cell stiffening in stiff hydrogels might contribute to the decreased motility of the cells. However, at this stage, it is not clear whether there is a functional relationship between the seen cell mechanical changes and the invasive behavior of cells.

As mentioned before, an important aspect of stiff tumor microenvironments is compressive stress [9]. This compressive stress is in our experimental system related to controllable parameters such as the hydrogel’s elastic properties and degradability as well as biological properties such as multicellularity. Several studies have analyzed the effect of compressive stress on the growth of tumor spheroids, e.g., spheroids growing in agarose gels or free-floating spheroids exposed to increased osmotic pressure [59,60,61]. So far, the effect of compressive stress on the mechanical properties of tumor spheroids has been not addressed in the context of matrix stiffness and degradability. Since we have found higher Brillouin frequency shifts for scenarios where compressive stress was increased (gel embedded versus single or free-floating spheroids), we suspect that the increased compaction and/or cellular stiffness is at least partly a response to compressive stress. This compressive stress can vary within the spheroid cross-section, as recently shown by compression sensors incorporated into aggregated spheroids that were then set under compressive stress [59]. The observation of increased Brillouin frequency shift values towards the spheroid center would also be in line with increased compressive stress within the center and increased spheroid compaction or the cells’ adaption of their mechanical properties as a response to it [59].

## 5. Conclusions

Taken together, our study brings new insights into the mechanical interplay between tumor spheroids and their complex 3D microenvironment. We have shown that the level of compressive stress varies in our system with matrix stiffness and degradability and modulates growth, morphology, invasion and mechanics of tumor spheroids. Increased elastic moduli of spheroids were associated with lower invasion into the matrix, which suggests that the cancer cells’ adaption of their mechanical properties in response to their microenvironments might also play a functional role in cancer progression. Since all these aspects can be modified with our experimental setup, we have a powerful model at hand to next investigate molecular mechanisms regulating the seen cellular responses. Ultimately, more insights into these mechanisms may provide a basis for identifying new targets that could potentially be used to interfere with tumor progression.

## Figures and Tables

**Figure 1 cancers-13-05549-f001:**
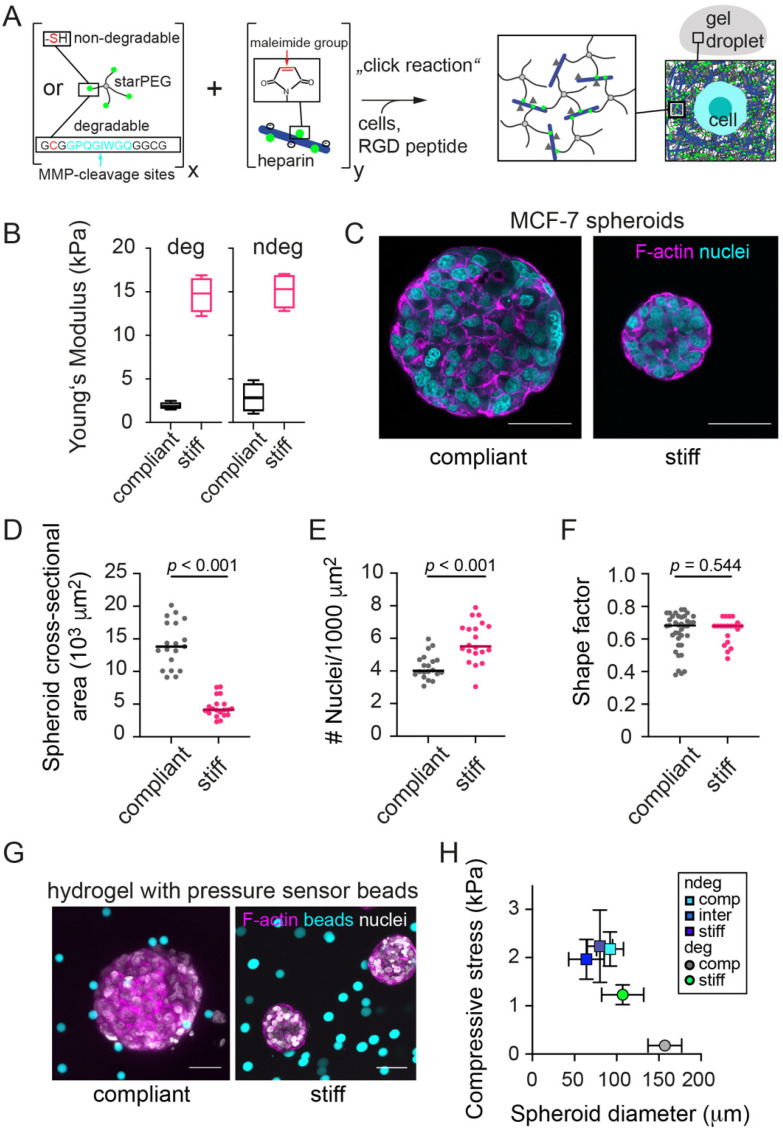
Hydrogel stiffness reduces the growth of MCF-7 spheroids. (**A**) Schematic of the PEG-heparin hydrogel system, into which initially single cells were embedded. (**B**) Elastic (Young’s) modulus for compliant and stiff hydrogels determined by AFM. *n* = 5–7 gels. (**C**) Representative confocal microscopy images of MCF-7 spheroids formed in a compliant and stiff degradable gel. Spheroid cultures were stained for F-actin and nuclei using Phalloidin-TRITC and DAPI respectively. Scatter plots showing (**D**) Cross-sectional area of MCF-7 spheroids in compliant and stiff degradable gels. Medians are indicated by a line. *n* = 20 spheroids each. (**E**) Number density of cells per unit area of cross-section of MCF-7 spheroids in compliant and stiff degradable hydrogels. *n* = 20 spheroids each. (**F**) Shape factors of tumor spheroids. (**D**–**F**) Shapes, sizes and densities were determined from confocal microscopy images of Phalloidin-TRITC/DAPI stained spheroid cultures using FIJI. A Mann–Whitney test was performed for statistical analysis (*p*-values are given). Spheroids analyzed after 14 days of culture. (**G**) Representative confocal microscopy images of MCF-7 spheroids that had grown together with elastic polyacrylamide beads in a compliant and stiff degradable gel as described in [30]. Spheroid cultures were stained for F-actin and nuclei using Phalloidin-TRITC and DAPI respectively; beads were fluorescently conjugated. (**H**) Compressive stress estimated from bead deformations as a function of the final spheroid diameter reached on day 14. Means +/− SEM are shown. All scale bars correspond to 50 µm.

**Figure 2 cancers-13-05549-f002:**
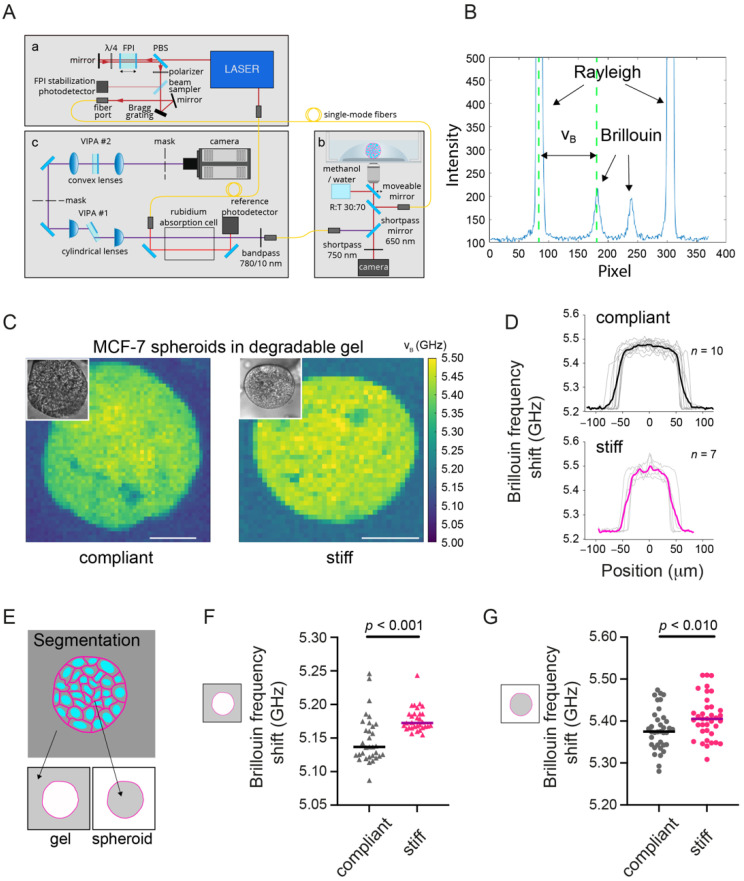
MCF-7 spheroids in stiff degradable hydrogels show an increased Brillouin frequency shift. (**A**) Combined confocal Brillouin microscopy setup. It consists of (a) illumination source, (b) confocal microscope, (c) Brillouin spectrometer. Abbreviations: FPI = Fabry–Pérot interferometer, PBS = polarizing beam splitter, VIPA = virtually imaged phased array. Adapted from Bakhshandeh et al., 2021. (**B**) Spectrum for a sample obtained from the setup consisting of Rayleigh peaks and Brillouin peaks as highlighted. (**C**) Representative Brillouin maps of MCF-7 spheroids grown in a compliant and stiff degradable PEG heparin hydrogel. Insets are brightfield images of the mapped spheroids. Scale bars correspond to 50 µm. (**D**) Line profiles of the Brillouin frequency shift across spheroids grown in compliant and stiff hydrogels. (**E**) Brillouin maps were segmented for hydrogel and spheroid area. (**F**) Scatter plot showing Brillouin frequency shifts for compliant and stiff degradable hydrogels. *n* = 31–32 spheroids each. Medians are indicated by a line. (**G**) Brillouin frequency shifts for MCF-7 spheroids in compliant and stiff degradable gels. *n* = 36–37 spheroids each. A Mann–Whitney test was done for statistical analysis (*p*-values are given). Spheroids analyzed after 14 days of culture.

**Figure 3 cancers-13-05549-f003:**
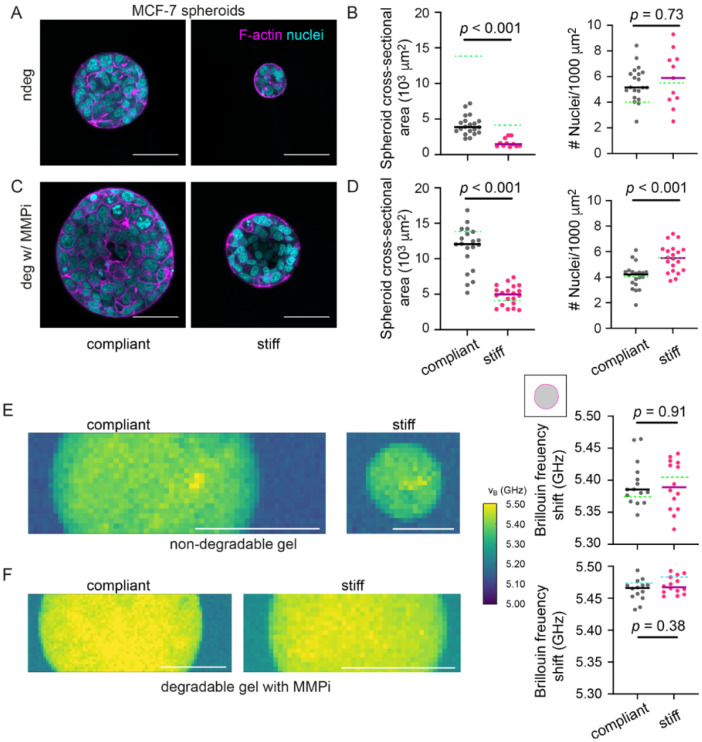
Blocking hydrogel degradation causes similar Brillouin frequency shifts between spheroids in compliant and stiff hydrogels. (**A**) Representative confocal microscopy images of MCF-7 spheroids grown in non-degradable compliant and stiff hydrogels. Spheroid cultures were stained for F-actin and nuclei using Phalloidin-TRITC and DAPI respectively. (**B**) Cross-sectional area and number density of cells per unit area of MCF-7 spheroids in compliant and stiff non-degradable gels. *n* = 10–20 spheroids each. (**C**) Representative confocal microscopy images of MCF-7 spheroids formed in a compliant and stiff degradable gel treated with 20 µM MMPi (**D**) Cross-sectional area and number density of cells per unit area of MCF-7 spheroids grown in compliant and stiff degradable gels treated with MMPi. *n* = 20 spheroids each. (**E**) Representative Brillouin maps of MCF-7 spheroids formed in a compliant and stiff non-degradable gel. Right: Scatter plot showing Brillouin frequency shifts for MCF-7 spheroids in compliant and stiff non-degradable gels. *n* = 14–15 spheroids each. Solid lines indicate medians. (**F**) Representative Brillouin maps of MCF-7 spheroids formed in a compliant and stiff degradable gel treated with MMPi. Right: Scatter plots showing Brillouin frequency shifts for MCF-7 spheroids in compliant and stiff degradable gels treated with MMPi. *n* = 14 spheroids each. Solid lines indicate medians. Dotted green and turquoise lines indicate medians of MCF-7 spheroids in (**B**,**E**) degradable gels or in (**D**,**F**) degradable gels treated with DMSO (vehicle control), respectively. A Mann–Whitney test was done for statistical analysis. *p*-values are given. All spheroids were analyzed after 14 days of culture. All scale bars correspond to 50 µm.

**Figure 4 cancers-13-05549-f004:**
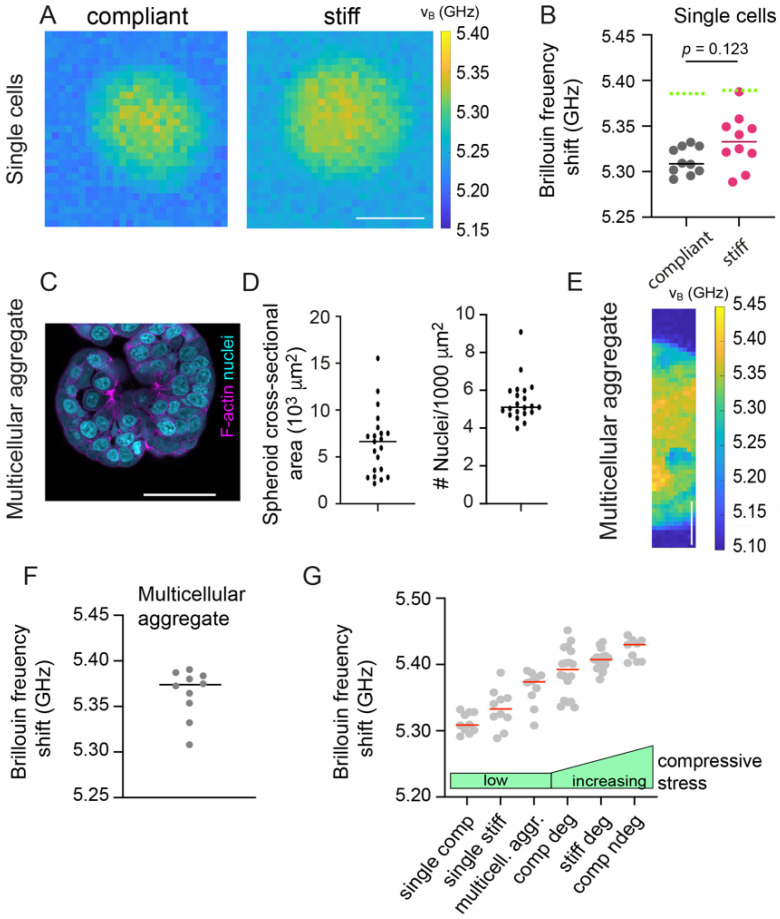
Brillouin frequency shifts increase with compressive stress. (**A**) Representative Brillouin maps of single MCF-7 cells grown in non-degradable compliant and stiff hydrogels at culture day 1. Scale bar corresponds to 10 µm. (**B**) Scatter plot showing corresponding Brillouin frequency shifts. *n* = 10. Solid lines indicate medians. Dotted green lines indicate medians of MCF-7 spheroids in non-degradable gels. A Mann–Whitney test was done for statistical analysis. *p*-values are given. (**C**) Representative confocal microscopy image of multicellular aggregates (not embedded within hydrogels) formed by MCF-7 cells 48 h after assembly. F-actin and nuclei were visualized using Phalloidin-TRITC and DAPI, respectively. Scale bar corresponds to 50 µm. (**D**) Cross-sectional area and number density of cells per unit area of multicellular MCF-7 aggregates. *n* = 20 spheroids. (**E**) Representative Brillouin map of multicellular aggregates as in C. (**F**) Scatter plot showing corresponding median Brillouin frequency shifts take from maps of multicellular aggregates as in E. *n* = 10. Line indicates median. Scale bar corresponds to 20 µm. (**G**) Median Brillouin frequency shifts obtained from a comparable set of experiments. For the different conditions, different levels of compressive stress level are expected as indicated by green bars, from single cells and free-floating multicellular aggregates to compliant and stiff degradable and non-degradable hydrogel cultures.

**Figure 5 cancers-13-05549-f005:**
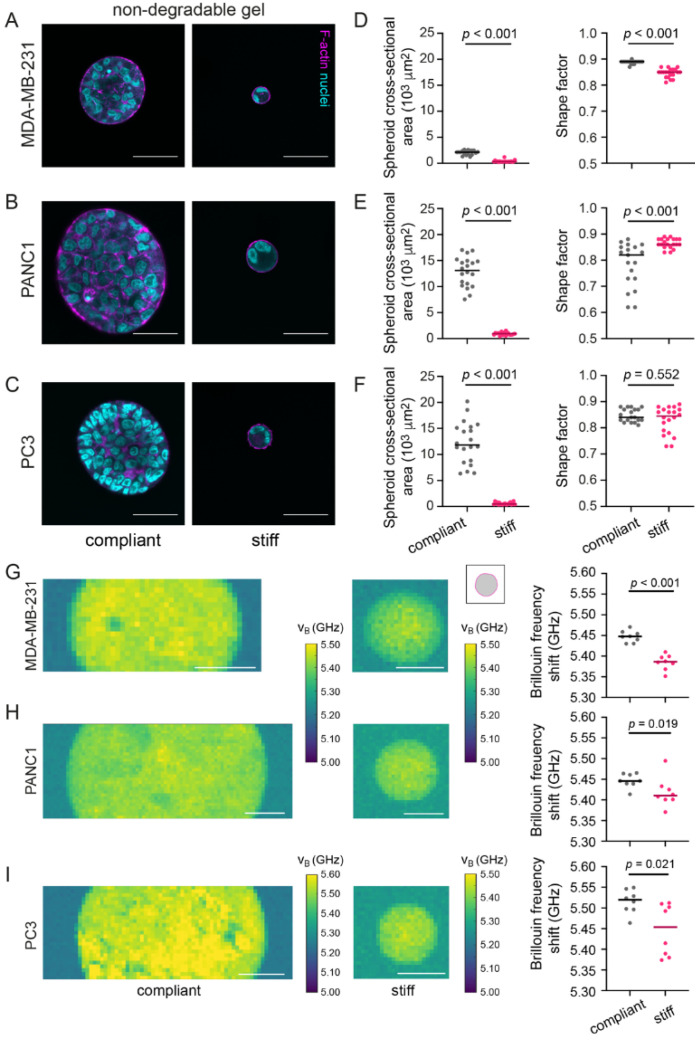
Invasive cancer cell lines (MDA-MB-231, PANC1 and PC3) do not form larger multicellular tumor spheroids in stiff non-degradable hydrogels and show lower Brillouin frequency shifts. (**A**–**C**) Representative confocal images of MDA-MB-231 (**A**), PANC1 (**B**) and PC3 (**C**) spheroids grown in a compliant and stiff non-degradable gel. Spheroid cultures were stained for F-actin and nuclei using Phalloidin-TRITC and DAPI respectively. Scale bars correspond to 50 µm. (**D**–**F**) Scatter plots showing cross-sectional area and number density of cells per unit area of respective spheroids in compliant and stiff non-degradable gels for the 3 cell lines on the left. *n* = 20 spheroids each. Lines indicate medians. (**G**–**I**) Representative Brillouin maps of MDA-MB-231 (**G**), PANC1 (**H**) and PC3 (**I**) spheroids formed in a compliant and stiff non-degradable gel (day 14). Scale bars correspond to 25 µm. Right: Scatter plots showing Brillouin frequency shifts for respective cell types. Lines indicate medians. *n* = 8 spheroids each. A Mann–Whitney test was done for statistical analysis. *p*-values are given. Spheroids analyzed after 14 days of culture.

**Figure 6 cancers-13-05549-f006:**
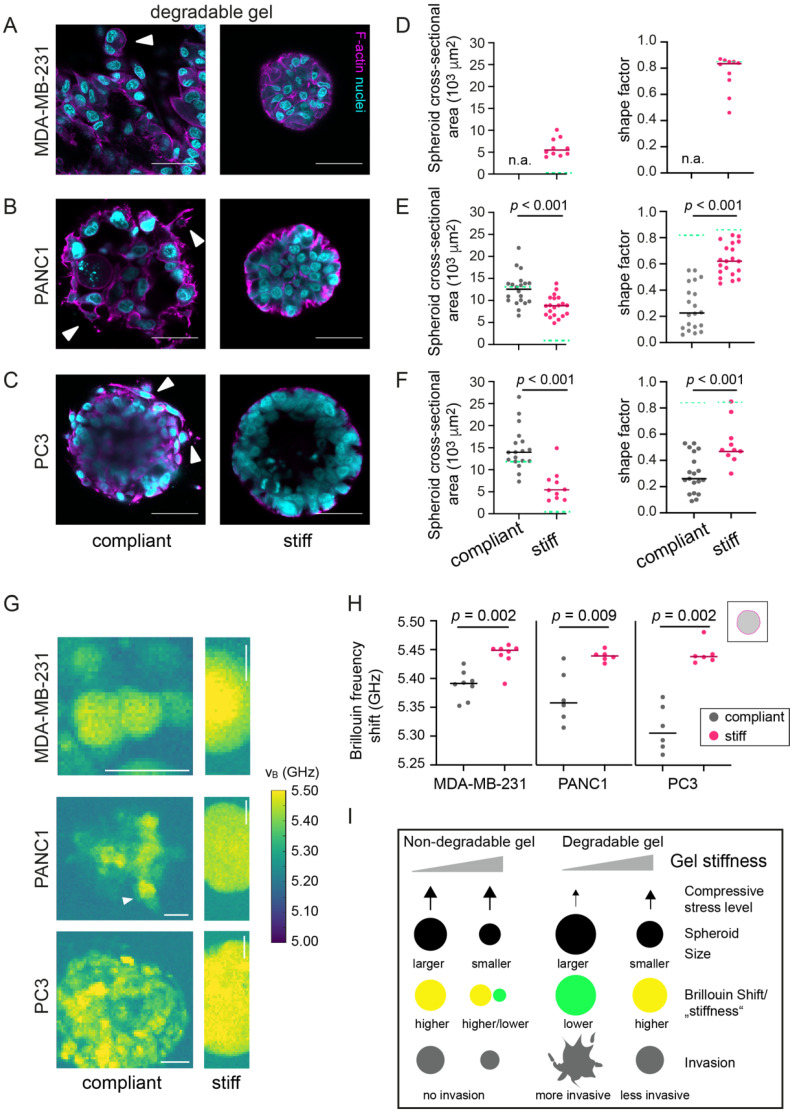
Invasive cancer cell lines (MDA-MB-231, PANC1 and PC3) show lower Brillouin frequency shift and invade into compliant degradable hydrogels. (**A**–**C**) Representative confocal microscopy images of MDA-MB-231 (**A**), PANC1 (**B**) and PC3 (**C**) spheroids grown in a compliant and stiff degradable gel. Scale bars correspond to 50 µm. (**D**–**F**): Scatter plots showing cross-sectional area and number density of cells per unit area of respective spheroids in compliant and stiff degradable gels. *n* = 20 spheroids each. Solid lines indicate medians. Dotted green lines indicate medians of respective spheroids in non-degradable gels. (**G**) Representative Brillouin maps of MDA-MB-231, PANC1 and PC3 spheroids formed in a compliant and stiff degradable gel. Scale bars correspond to 25 µm. (**H**) Scatter plots showing Brillouin frequency shifts for respective cell types. Lines indicate medians. *n* = 6–8 spheroids each. A Mann–Whitney test was done for statistical analysis. *p*-values are given. Spheroids analyzed after 14 days of culture. (**I**) Summary of results on compressive stress, spheroid growth, mechanics and invasion in dependence of matrix degradability and stiffness.

## Data Availability

Please refer to Appendix A.

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
