# Peer review of "Mapping Tumor Spheroid Mechanics in Dependence of 3D Microenvironment Stiffness and Degradability by Brillouin Microscopy"

_cancers, 2021, doi:10.3390/cancers13215549_

Round 1

Reviewer 1 Report

This is a very well written manuscript describing the in-situ measurement of stiffness across multicellular spheroids embedded within mechanically different hydrogels. The work primarily establishes the application of Brillouin Microscopy (BM) in in-situ stiffness characterization of single cell and multicellular spheroid mechanical properties. It also shows how these stiffness values change in response to the environment the cells are embedded in. The measured differences in mechanical properties of multicellular tumor cell spheroids grown within compliant or stiff, degradable or non-degradable hydrogels match prior measurements done using other more destructive techniques. These results attest the applicability of BM for in-situ stiffness measurement purposes. Secondary measurements of compressive stresses in these gels also indicate potential mechanisms that drive the changes in cellular and spheroid mechanical properties. There are a few minor questions and suggestions that, if addressed could make the manuscript better - 

1) Caption for figure 4 (A and B especially) does not seem the match the figures

2) In figure 6 I, the Brillouin shift for cells in non-deg stiff gels should read lower, not higher. 

3) Highly invasive cancer cells seem to be softer in stiffer, non-degradable gels, but stiffer in stiff, degradable cells. This observation could be further elaborated in the discussion section beyond what is mentioned in lined 537-539. 

4) A common mechanosensing pathway and mechanism for mechanical adaption of cells to the environment relies on the formation of cell-substrate bonds. I think, the fact that the substrate is a PEG based hydrogel, the cells in this case do not interact with the substrate and mechanosensation is primarily driven by pressure or stress sensing. Is this correct? And if so, discussing the implications of this in the manuscript will help the reader. 

5) There is only a brief mention about the role of intracellular density and water content on BM observations. However, I wonder if that plays a much more significant role here where the external hydrostatic pressure due to hydrogel compression is balanced by an increased osmotic pressure inside individual cells and the spheroids as a whole, which increases the intracellular density/decreases water content and consequently the increased Brillouin shift. There might be an obvious reason why this is not the case that I am unaware of, but if the authors could clarify, that would help. 

Once again, these are minor comments and overall this is an excellent manuscript. 

Reviewer 2 Report

Brillouin microscopy is a noncontact method but I am still interested whether they were any studies done that report any influences of such high frequencies on cells?

With Brillouin microscopy in principal you get an information about the compressibility which is heavily dependent on the water content, which you also showed in your experiments. The cells were embedded in the gel spheroids and as you wrote it is hard to distinguish  whether the frequency shifts are due to changed mechanical properties of single cells, clusters of cells or due to changes in the whole systems (cells plus hydrogel). Have you any ideas how the influence of individual components could be distinguished?
